# Propensity score methods for comparative-effectiveness analysis: A case study of direct oral anticoagulants in the atrial fibrillation population

**Giorgio Ciminata**[1]*, **Claudia Geue**[1], **Olivia Wu**[1], **Manuela Deidda**[1], **Noemi Kreif**[2], **Peter Langhorne**[3]

**1** Health Economics and Health Technology Assessment (HEHTA), Institute of Health & Wellbeing, University of Glasgow, Glasgow, United Kingdom, **2** Centre for Health Economics, University of York, York, United Kingdom, **3** Institute of Cardiovascular & Medical Sciences, University of Glasgow, Glasgow, United Kingdom

* Giorgio.Ciminata@glasgow.ac.uk

**Data Availability Statement:** Data can only be obtained through application to Information Services Division (ISD) via the Public Benefit and Privacy Panel (PBPP). Due to legal restrictions, Data can only be accessed by the researchers

## Abstract

### Objective

To explore methodological challenges when using real-world evidence (RWE) to estimate comparative-effectiveness in the context of Health Technology Assessment of direct oral anticoagulants (DOACs) in Scotland.

### Methods

We used linkage data from the Prescribing Information System (PIS), Scottish Morbidity Records (SMR) and mortality records for newly anticoagulated patients to explore methodological challenges in the use of Propensity score (PS) matching, Inverse Probability Weighting (IPW) and covariate adjustment with PS. Model performance was assessed by standardised difference. Clinical outcomes (stroke and major bleeding) and mortality were compared for all DOACs (including apixaban, dabigatran and rivaroxaban) versus warfarin. Patients were followed for 2 years from first oral anticoagulant prescription to first clinical event or death. Censoring was applied for treatment switching or discontinuation.

### Results

Overall, a good balance of patients' covariates was obtained with every PS model tested. IPW was found to be the best performing method in assessing covariate balance when applied to subgroups with relatively large sample sizes (combined-DOACs versus warfarin).

With the IPTW-IPCW approach, the treatment effect tends to be larger, but still in line with the treatment effect estimated using other PS methods. Covariate adjustment with PS in the outcome model performed well when applied to subgroups with smaller sample sizes (dabigatran versus warfarin), as this method does not require further reduction of sample size, and trimming or truncation of extreme weights.

named in the PBPP application and via the National-Safe-Haven platform, a secure environment used to maintain the privacy and confidentiality of the personal information held. Therefore, Data cannot be exported or shared. Here is the link of the Public Benefit and Privacy Panel (PBPP), with details on how to request access to health data in Scotland. https://www.informationgovernance.scot.nhs.uk/pbpphsc/.

**Funding:** This work was supported by Farr Institute for Health Informatics Research, Scotland, MRC, grant number MR/K007017/1. The funders had no role in study design, data collection and analysis, decision to publish, or preparation of the manuscript.

**Competing interests:** GC and CG have received research grants from Bristol-Myers Squibb UK and Pfizer UK outside the submitted work. OW has received consulting fee from Bayer UK outside the submitted work. MD, NK and PL declare no conflict of interest.

## Conclusion

The choice of adequate PS methods may vary according to the characteristics of the data. If assumptions of unobserved confounding hold, multiple approaches should be identified and tested. PS based methods can be implemented using routinely collected linked data, thus supporting Health Technology decision-making.

## Introduction

Comparative-effectiveness research aims to reduce the gap between clinical research and clinical practice [1, 2], thus, providing clinicians, patients and policy makers with the clinical evidence needed to make informed decisions concerning healthcare. In this context, both randomised controlled trials (RCTs) and real-world evidence (RWE) contribute to generating clinical evidence for decision-making.

In randomised trials, randomisation ensures that differences in patient characteristics such as age, sex, comorbidities and disease severity, are similarly distributed between treatment groups; the observed difference in term of outcome between the treatment groups in the study population can be attributable to the treatment [2].

In RWE, the absence of randomisation does not allow for an unbiased comparison between patients who are exposed and those who are not exposed to the treatment under study. Hence, the observed differences in health outcomes between the groups may be influenced by the population characteristics or other additional factors rather than by the treatment.

Crucially, a lack of randomisation in RWE studies gives rise to confounding by indication, occurring when the prognostic factors, such as disease severity, used for treatment selection also affect the outcome [1]. For instance, patients with more severe conditions receive more intense treatments, and as a result, when comparing outcomes among treatment groups in a naïve way, the more intensive treatment may be associated with poorer outcomes. Nevertheless, RWE may provide additional insights concerning safety and effectiveness of a treatment and in some cases may be the only available source of evidence if randomised data are not available [1].

Thus, RWE is increasingly used in Health Technology Assessment to inform reimbursement and coverage decisions. In this context, RWE is used in the "accelerated market access" process where initial decisions are conditional on additional randomised and non-randomised evidence generated over time [3].

Historically, regression adjustment has been used to address confounding in RWE; but over the last decade, there has been an increasing interest in the application of Propensity Score based (PS) methods, such as matching and inverse probability weighting (IPW), when using observational data in medical research. Propensity score methods attempt to mimic the process or randomisation by estimating the probability of treatment assignment conditional on observed baseline characteristics [1, 4].

Propensity score methods offer several advantages over conventional regression methods [4]. However, while PS methods may reduce the bias due to observable confounders such as age, sex and existing comorbidities, other unobserved confounding, such as patients' tolerability and access to healthcare may still bias the PS estimates. Propensity score methods can address observed confounding if the assumption of 'no unobserved confounding' is reasonable, i.e. that the investigator was able to measure al variables that both influence the treatment assignment and are prognostic of the outcome [5]. To account for the presence of both

observed and unobserved confounding, different statistical methods such as instrumental variable, difference in differences and regression discontinuity should be used as an alternative to PS methods [6, 7].

The objective of this study is to explore methodological challenges in using RWE, with a focus on PS based methods, to estimate comparative-effectiveness for a case study of direct oral anticoagulants (DOACs); a class of drugs, including apixaban, dabigatran and rivaroxaban, used for the prevention of stroke in the population affected by atrial fibrillation (AF). The rapid onset of action, following oral administration, is one of the main assets of DOACs. The predictability of pharmacodynamics and pharmacokinetics allows DOACs to be used at a fixed dose without requirement for routine anticoagulation monitoring [8, 9]. This study forms part of a wider project that used routinely collected data where clinical and comparative-effectiveness of DOACs are assessed in greater detail.

Confounding by indication appears to be an issue with DOACs, and most studies assessing their comparative- effectiveness have used different PS based methods to address observed confounders [10, 11]. However, these studies often do not provide a rationale for choosing a specific PS based method, among different variations. In most studies assessing the effectiveness of DOACs (either head-to-head or compared to warfarin), neither the comparison between PS methods nor the reason for selecting a specific PS method are provided [10, 11].

Despite the fact that there are clear differences between PS methods, the choice of one method over another often appears to be arbitrary without a clear rationale supporting model selection. Among twenty-two studies identified from a recent systematic review accessing the effectiveness of DOACs compared to warfarin [11], only two studies appeared to compare PS methods or justify PS model selection.

## Methods

### Data sources and cohort

Fully anonymised data were obtained from the Information Services Division (ISD) of NHS Scotland as part of a wider project that used routinely collected data to evaluate the comparative-effectiveness of DOACs in the prevention of stroke in the AF population. Scotland offers a robust record linkage system, where administrative patient-level health data are routinely collected.

All patients treated with either warfarin or DOACs between 2009 and 2017 were identified from the Scottish Prescribing Information System (PIS), a database that includes prescribing records for all medicines and their associated costs, which are prescribed and dispensed by community pharmacies, dispensing doctors, and a small number of specialist appliance suppliers [12]. Records from PIS are available from 2009 onwards; therefore, to establish a cohort of patients with a first prescription of warfarin or DOACs, and no exposure to anticoagulation within one year prior to the index date, only patients starting anticoagulation from 2010 onwards were included in the analysis. Individual-level data linkage was then carried out with General Acute Inpatient and Day Case Scottish Morbidity Records 01 and mortality records to identify a cohort of AF patients (defined using ICD-10 code I48X) and clinical and mortality events. Inpatient records contain all general acute admissions, categorized as inpatient stays or day cases, discharged from non-obstetric and non-psychiatric specialties [13]. The clinical outcomes were identified from SMR01 according to ICD-10 and OPCS-4 codes (S1 Table).

Further, to ensure that only patients that were likely to have received OACs because of an AF diagnosis were included, any patients with a diagnosis other than AF were excluded from the analysis. Clinical codes for inclusion and exclusion criteria are presented in the S2 Table.

From our cohort of AF patients who are first time OAC users, we defined three subgroups: those on warfarin, those on any DOACs (including prescriptions of apixaban, dabigatran and rivaroxaban), and those on dabigatran only. We included any DOACs prescription into a single subgroup to assure an adequate overall treatment sample size; we refer to this subgroup as the combined-DOACs subgroup. The subgroup including only AF patients on dabigatran, is the smallest subgroup of DOACs users in Scotland compared to apixaban and rivaroxaban [14] and was therefore used to assess whether any of the PS approaches tested was sensitive to sample size. Thus, two comparisons were possible: combined-DOACs versus warfarin and dabigatran versus warfarin.

## Propensity score estimation

Propensity score methods estimate the probability of treatment assignment conditional on observed baseline characteristics [1, 4]. The PS estimation was carried out for each of our subgroups (warfarin, the combined-DOACs and dabigatran), resulting in two different PS models.

Propensity scores were estimated with a logit model accounting for a series of baseline characteristics of first time OAC users. We accounted for age and sex, which are relevant drivers of treatment choice and are prognostic of the outcomes of interest.

We also accounted for socio economic status using the Scottish Index of Multiple Deprivation (SIMD), reflecting areas of multiple deprivation ranked from the most to the least deprived and measured as quintiles, where the most and the least deprived areas are represented by 1 and 5, respectively [15].

Further, PS were estimated taking into account the risk score calculated (for each patient for the 5-year period prior to their first anticoagulation prescription) with the risk prediction tools $CHA_2DS_2$-VASc and HAS-BLED designed to stratify respectively in the context of AF the risk of stroke and the risk of bleeding [16–18]. Other relevant confounders that we used in our PS estimation were ischaemic stroke or systemic embolism or transient ischaemic attack (TIA), vascular disease, hypertension, diabetes, cancer, prescription predisposing bleeding, and comorbidity. The PS in each of the two different PS models was estimated according to the full set of covariates listed above. The proportion of missing data was <5%, hence imputation of missing values was not used [19], and a complete available case analysis was carried out.

For each PS model tested, and for each comparison (combined-DOACs versus warfarin and dabigatran versus warfarin) PS distribution was inspected graphically to identify potentially extreme weights and to ascertain whether an adequate overlapping distribution has been achieved. Extreme weights are considered as such if PSs are <0.1 for the treatment group (combined-DOACs and dabigatran) or >0.9 for the control group (warfarin). Distributions of the predicted probabilities between treatment and control groups should overlap to indicate that covariates between groups are comparable [4, 20].

## Propensity score models assessed

Assuming that the assumption of no unobserved confounding was reasonable, and with the support of guidelines on the use of observational data to inform estimates of treatment effectiveness in technology appraisal [5], for the combined-DOACs versus warfarin and dabigatran versus warfarin comparisons, we identified and tested different PS based methods: PS matching, covariate adjustment including PS as covariate and a series of IPW methods.

With the propensity score matching we created a sub-sample of each treatment group, and for each comparison, sharing a similar PS value. This allows outcomes between treatment groups to be directly compared [4].

A key aspect of this PS method is whether matching should be done with or without replacement. In the first case, any patient from the control group can be used several times for more than one treated individual. Replacement is particularly useful in settings where the treatment group significantly outnumbers the control group. By contrast, matching without replacement allows patients from the control group to be matched against those in treatment group only once [4, 21]. In our data, warfarin (the control) outnumbers the treatment group (combined-DOACs and dabigatran); this is due the adoption of DOACs for the prevention of stroke in the AF population being relatively recent compared to warfarin [22]. Thus, PS matching without replacement was selected as the most suitable PS matching method. In the covariate adjustment method, the only steps required in the PS model were deciding the functional form of the regression model and PS estimation.

The IPW methods we tested were Inverse Probability of Treatment Weighting (IPTW) and IPTW combined with Inverse Probability of Censoring Weighting (IPCW). In the IPTW method, a weight, reflecting the probability of being exposed to either combined-DOACs or dabigatran and equal to the reciprocal of the PS, was assigned to each patient in the treatment group. A weight equal to the reciprocal of one minus the PS was assigned to patients in the warfarin group. In the IPW method combining IPTW with IPCW, two different sets of weights were estimated. The weights for IPTW were estimated as discussed. Those for IPCW were estimated by censoring patients who switch treatment, and by assigning weights to individuals who were not censored but shared similar characteristics with the switchers.

Then, IPTW and IPCW weights were multiplied to obtain the overall weight reflecting ATE and censoring [4, 23, 24].

The adequacy of model specification for PS matching and IPW methods was assessed by means of standardized differences; a measure generally used to compare the mean of variables between treatment and control groups. The use of standardised differences for balance assessment has been advocated in the literature as it is invariant to sample size and can be applied across different PS methods. Further, such measure is easily interpretable using graphical displays even with a large number of covariates [4, 20]. For the covariate adjustment method, we have used "weighted conditional" standardised difference as described by Austin (2008). With this method, the pooled standard deviation, obtained from the difference in the mean of a covariate between treated and untreated subjects, is integrated over the distribution of the propensity score [20].

With both methods, differences in the means of covariates is considered negligible if below the threshold of 0.1 standard deviation [25]. Although there is no universal agreement on what the threshold for the standardised difference should be, the threshold of 0.1 is now considered by researchers as an adequate measure for diagnostic purposes assessing covariates balance and imbalance [26].

The PS methods described above have been used to estimate the Average treatment effect (ATE), defined as the average treatment effect for the entire population (i.e. regardless of whether a particular individual has been treated) [27]. Specifically, in our analysis, the ATE, being the estimand of interest, was estimated on the whole AF population and for each comparison i.e., ATE of being treated with any DOAC (combined-DOACs) and ATE of being treated with dabigatran.

## Outcome model

Cox proportional hazards regression was used to compare risks between control and treatment groups, for each comparison and for three major AF related clinical events: stroke-all (including haemorrhagic and ischaemic), major bleeding and all-cause mortality. To compensate for any potential remaining covariate imbalance and further reducing the bias caused by residual

differences in observed baseline covariates, we included age, sex, socio-economic status and comorbidity in our outcome models. The other variables, described in the *Propensity Score Estimation* subsection, were assumed to be captured by comorbidity and were, therefore not included. Patients were censored if they switched or discontinued treatment; for each method, the risks of stroke, major bleeding or death (for patients exposed to either DOACs or warfarin) were estimated from anticoagulation initiation to the time of clinical event or death during a 2-year follow-up period.

The first clinical event for each treatment was determined within a competing risk framework. In this analysis, treatment discontinuation, i.e. temporal gaps between consecutive prescriptions, was considered to be occurring if the gap exceeded a 28 days threshold, and the supply of the penultimate prescription did not fill the gap.

The threshold was identified in a drug utilisation study using the same patient-level data utilised in this paper [22]. For the IPW method combining IPTW with IPCW, censoring was specifically modelled in the PS model.

As previously described, patients who switched treatment were censored; while weights were assigned to individuals who were not censored but shared similar characteristics with the switchers. These weights were then multiplied by the weights obtained from IPTW.

In addition to comparing PS models in terms of performance by measuring the standardised differences for each covariate, the ATE, estimated with the outcome model for each of the clinical outcomes selected, was compared across method to assess whether and how it differs depending on the PS method used.

## Ethics and data sharing

We obtained the necessary permissions and approvals to access these national datasets. No ethical approval was needed. All data underlying the analyses are confidential and subject to disclosure control. Data can only be obtained through application to Information Services Division (ISD) via the Public Benefit and Privacy Panel (PBPP).

## Results

### Cohort characteristics

From the cohort of first time OAC users identified from the PIS between 2009 and December 2017, two subgroups of patients on either warfarin (34,876) or combined-DOACs (15,142) were identified. Among the combined-DOACs users, 622 patients were on dabigatran. Overall, mean age of patients at the time of the first prescription was similar across all treatment groups.

Across all treatments, patients with the highest risk of stroke and the lowest risk of bleeding, measured using the CHA2DS2-VASc and HAS-BLED score respectively, represented the majority. While most patients had no comorbidities across all treatment groups, those on warfarin represented the biggest proportion. Further, the proportion of patients with a history of stroke or TIA was lower in the warfarin group than any other treatment group. About one third of patients on anticoagulation had hypertension, which is an important risk factor for stroke. In addition, the majority of patients across all treatment groups were also on medication predisposing to bleeding such as aspirin and non-steroidal anti-inflammatory drugs. Patients' baseline characteristics are reported in Table 1.

### Propensity score distribution

The PSs for the combined-DOACs versus warfarin comparison showed an adequate overlapping distribution, (Fig 1A). This was also observed in the dabigatran versus warfarin

**Table 1. Baseline characteristics.**

| Characteristics | Warfarin | Combined-DOACs | Dabigatran |
|---|---|---|---|
| | N (%) | N (%) | N (%) |
| Subgroup | 34,876 | 15,142 | 622 |
| Sex | | | |
| Men | 20,007 (57.37) | 8,433 (55.69) | 378 (60.77) |
| Women | 14,869 (42.63) | 6,709 (44.31) | 244 (39.23) |
| Mean age (SD) | 75 (11.09) | 74(11.32) | 72(11.10) |
| SIMD (Scottish index of multiple deprivation) | | | |
| 1 (most deprived) | 6,814 (19.54) | 2,813 (18.58) | 87 (13.99) |
| 2 | 7,420 (21.28) | 2,965 (19.58) | 104 (16.72) |
| 3 | 7,297 (20.92) | 3,039 (20.07) | 149 (23.95) |
| 4 | 6,977 (20.01) | 3,110 (20.54) | 171 (27.49) |
| 5 (least deprived) | 6,368 (18.26) | 3,215 (21.23) | 111 (17.85) |
| CHA2DS2-VASc score (risk of stroke) | | | |
| 0–1 (low to moderate risk) | 7,705 (22.09) | 3,429 (22.65) | 171 (27.49) |
| 2–3 (moderate to high risk) | 11,232 (32.21) | 4,606 (30.42) | 195 (31.35) |
| ≥4 (high risk) | 15,939 (45.70) | 7,107 (46.94) | 256 (41.16) |
| HAS-BLED score (risk of bleeding) | | | |
| 0–2 (low to moderate risk) | 24,875 (71.32) | 9,862 (65.13) | 447 (71.86) |
| ≥3 (moderate to high risk) | 10,001 (28.68) | 5,280 (34.87) | 175 (28.14) |
| Comorbidity | | | |
| No comorbidity | 18,374 (52.68) | 6,502 (42.94) | 311 (50.00) |
| 1 comorbidity | 6,952 (19.93) | 3,525 (23.28) | 133 (21.38) |
| >1 comorbidity | 9,550 (27.38) | 5,115 (33.78) | 178 (28.62) |
| Stroke or TIA | 2,542 (7.29) | 1,912 (12.63) | 80 (12.86) |
| Vascular disease | 4,903 (14.06) | 2,562 (16.92) | 85 (13.67) |
| Hypertension | 10,901 (31.26) | 5,361 (35.40) | 200 (32.15) |
| Diabetes mellitus | 4,449 (12.76) | 2,275 (15.02) | 85 (13.67) |
| Cancer | 2,904 (8.33) | 1,342 (8.86) | 43 (6.91) |
| Drugs causing bleeding | 18,843 (54.03) | 8,453 (55.82) | 314 (50.48) |

Note: DOACs = Direct Oral Anticoagulants, SIMD = Scottish Index of Multiple Deprivation, TIA = Transient Ischaemic Attack.

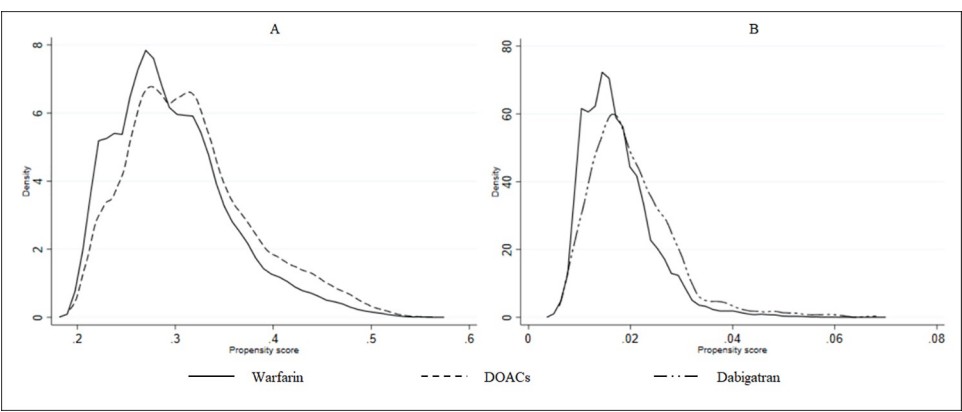

**Fig 1. Propensity score distribution for warfarin, combined-DOACs, and dabigatran.** Note: DOACs = Direct Oral Anticoagulants.

| Characteristics | Unadjusted | | PSM adjusted | | PS covariate adjusted | | IPW adjusted | |
|---|---|---|---|---|---|---|---|---|
| Women | 0.034 | • | 0.001 | • | 0.001 | • | 0.001 | • |
| Mean age (SD) | 0.037 | • | 0.025 | • | 0.011 | • | 0.004 | • |
| SIMD quintile 2 | 0.042 | • | 0.001 | • | 0.003 | • | 0.001 | • |
| SIMD quintile 3 | 0.021 | • | 0.001 | • | 0.001 | • | 0.001 | • |
| SIMD quintile 4 | 0.013 | • | 0.011 | • | 0.002 | • | 0.001 | • |
| SIMD quintile 5 | 0.075 | • | 0.008 | • | 0.016 | • | 0.001 | • |
| CHA2DS2-VASc score 2-3 | 0.039 | • | 0.004 | • | 0.001 | • | 0.001 | • |
| CHA2DS2-VASc score ≥4 | 0.025 | • | 0.026 | • | 0.002 | • | 0.001 | • |
| HAS-BLED score ≥3 | 0.133 | • | 0.002 | • | 0.018 | • | 0.004 | • |
| Comorbidity 1 comorbidity | 0.081 | • | 0.005 | • | 0.052 | • | 0.001 | • |
| Comorbidity >1 comorbidity | 0.139 | • | 0.021 | • | 0.055 | • | 0.001 | • |
| Stroke or TIA | 0.179 | • | 0.008 | • | 0.063 | • | 0.001 | • |
| Vascular disease | 0.079 | • | 0.029 | • | 0.066 | • | 0.001 | • |
| Hypertension | 0.088 | • | 0.006 | • | 0.016 | • | 0.004 | • |
| Diabetes mellitus | 0.066 | • | 0.023 | • | 0.060 | • | 0.001 | • |
| Cancer | 0.019 | • | 0.014 | • | 0.011 | • | 0.002 | • |
| Drugs causing bleeding | 0.036 | • | 0.007 | • | 0.002 | • | 0.001 | • |
| | 0.00 0.10 0.20 0.30 | | 0.00 0.10 0.20 0.30 | | 0.00 0.10 0.20 0.30 | | 0.00 0.10 0.20 0.30 | |

**Fig 2. Covariate imbalance assessment for combined-DOACs vs. warfarin.** Note: PS = Propensity Score, PSM = Propensity Score Matching, IPW = Inverse Probability Weighting, SIMD = Scottish Index of Multiple Deprivation, TIA = Transient Ischaemic Attack.

comparison; however, the totality of the PS generated were extreme and had a poor overlap (Fig 1B). In these cases, applying PS trimming or extreme weights truncation is clearly not feasible.

## Covariate balance assessment

Following the first graphical assessment on the PS models specification, the distribution of baseline covariates between treatment groups was assessed by means of standardized differences. As shown in Fig 2, the unadjusted standardized differences indicated an adequate starting balance for most of the baseline characteristics of patients on combined-DOACs or on warfarin, with differences in the means of covariates below the threshold of 0.1 standard deviation.

Overall, a good balance of patients' covariates was obtained with every PS model tested. However, the standardised difference for CHA2DS2-VASc score ≥4, did not improve with the PS matching method. Nevertheless, the standardised difference for these patient characteristics was still below the threshold regardless of being adjusted or unadjusted.

The standardized difference above the threshold reflected some differences in terms of age, socio-economic status and previous history of stroke or TIA between dabigatran and warfarin users in the starting baseline characteristics. However, an adequate balance was achieved for all covariates with the PS covariate adjustment method. Propensity score matching failed to provide a good balance in terms of patient characteristics between the dabigatran and warfarin groups. Similarly, improved balance was not achieved for every covariate when using the IPW approach. In particular, the balance for socio demographic characteristics captured by SIMD (category 5) and the covariate indicating a high risk of bleeding (HAS-BLED score ≥3) although still below the threshold, was suboptimal compared to the unadjusted initial baseline characteristics balance (Fig 3).

## Clinical outcomes

Despite the differences in terms of baseline characteristics and sample size, the treatment effect is comparable across methods (Figs 4 and 5). However, the treatment effect estimated with the IPTW-IPCW approach, tends to be larger, but still in line, compared to the treatment effect

| Characteristics | Unadjusted | | PSM adjusted | | PS covariate adjusted | | IPW adjusted | |
|---|---|---|---|---|---|---|---|---|
| Women | 0.069 | • | 0.036 | • | 0.006 | • | 0.016 | • |
| Mean age (SD) | 0.129 | • | 0.018 | • | 0.024 | • | 0.010 | • |
| SIMD quintile 2 | 0.116 | • | 0.017 | • | 0.009 | • | 0.005 | • |
| SIMD quintile 3 | 0.073 | • | 0.027 | • | 0.042 | • | 0.005 | • |
| SIMD quintile 4 | 0.177 | • | 0.007 | • | 0.057 | • | 0.011 | • |
| SIMD quintile 5 | 0.011 | • | 0.008 | • | 0.005 | • | 0.014 | • |
| CHA2DS2-VASc score 2-3 | 0.018 | • | 0.053 | • | 0.003 | • | 0.007 | • |
| CHA2DS2-VASc score ≥4 | 0.092 | • | 0.010 | • | 0.007 | • | 0.013 | • |
| HAS-BLED score ≥3 | 0.012 | • | 0.021 | • | 0.009 | • | 0.014 | • |
| Comorbidity 1 comorbidity | 0.036 | • | 0.008 | • | 0.003 | • | 0.002 | • |
| Comorbidity >1 comorbidity | 0.027 | • | 0.021 | • | 0.023 | • | 0.001 | • |
| Stroke or TIA | 0.186 | • | 0.024 | • | 0.026 | • | 0.002 | • |
| Vascular disease | 0.011 | • | 0.028 | • | 0.007 | • | 0.008 | • |
| Hypertension | 0.019 | • | 0.003 | • | 0.001 | • | 0.010 | • |
| Diabetes mellitus | 0.027 | • | 0.013 | • | 0.019 | • | 0.005 | • |
| Cancer | 0.053 | • | 0.019 | • | 0.021 | • | 0.016 | • |
| Drugs causing bleeding | 0.071 | • | 0.016 | • | 0.001 | • | 0.018 | • |
| | 0.00 0.10 0.20 0.30 | | 0.00 0.10 0.20 0.30 | | 0.00 0.10 0.20 0.30 | | 0.00 0.10 0.20 0.30 | |

**Fig 3. Covariate imbalance assessment for dabigatran vs. warfarin.** Note: PS = Propensity Score, PSM = Propensity Score Matching, IPW = Inverse Probability Weighting, SIMD = Scottish Index of Multiple Deprivation, TIA = Transient Ischaemic Attack.

estimated with other PS methods. This is particularly evident when the sample size of the treatment subgroup is relatively small (Fig 5).

## Discussion

In clinical practice, population case mix may diverge substantially, making a comparison of safety and effectiveness of two health interventions difficult. Propensity score methods allow

| Outcome | Method | HR 95% Conf. Interval | |
|---|---|---|---|
| **Stroke all** | | | |
| | Crude | 0.97 (0.85 - 1.11) | |
| | PSM | 0.96 (0.83 - 1.12) | |
| | PS covariate adjusted | 0.93 (0.81 - 1.06) | |
| | IPTW | 0.93 (0.82 - 1.07) | |
| | IPTW-IPCW | 0.95 (0.84 - 1.11) | |
| **Major bleeding** | | | |
| | Crude | 0.99 (0.90 - 1.08) | |
| | PSM | 1.00 (0.89 - 1.11) | |
| | PS covariate adjusted | 0.99 (0.90 - 1.08) | |
| | IPTW | 0.99 (0.90 - 1.09) | |
| | IPTW-IPCW | 1.01 (0.93 - 1.10) | |
| **Mortality-all cause** | | | |
| | Crude | 1.18 (1.11 - 1.26) | |
| | PSM | 1.21 (1.12 - 1.30) | |
| | PS covariate adjusted | 1.17 (1.10 - 1.25) | |
| | IPTW | 1.18 (1.11 - 1.26) | |
| | IPTW-IPCW | 1.18 (1.12 - 1.25) | |
| | | 0.25  0.50  1.00  2.00  4.00 | |
| | | Favouring DOACs    Favouring warfarin | |

**Fig 4. HRs for combined-DOACs vs. warfarin by propensity score methods.** Note: DOACs = Direct Oral Anticoagulants, PS = Propensity Score, PSM = Propensity Score Matching, IPTW = Inverse Probability of Treatment Weighting (IPTW), Inverse Probability of Censoring Weighting (IPCW).

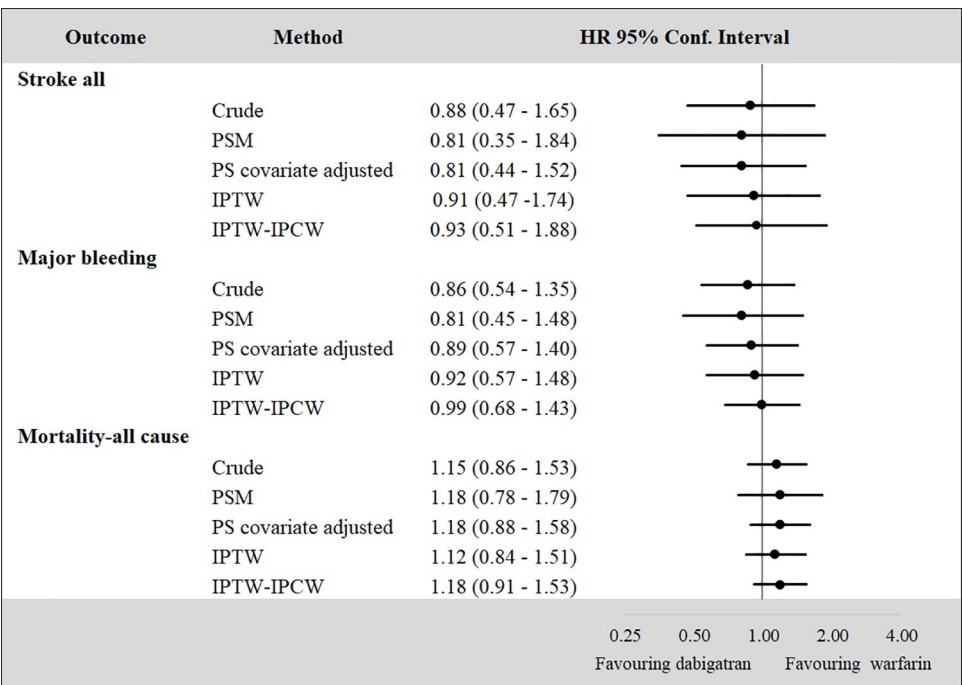

**Fig 5. HRs for dabigatran vs. warfarin by propensity score methods.** Note: PS = Propensity Score,
PSM = Propensity Score Matching, IPTW = Inverse Probability of Treatment Weighting (IPTW), Inverse Probability
of Censoring Weighting (IPCW).

for reducing any potential imbalance between covariates and obtaining more homogenous
and comparable treatment groups [4, 28].

Although PS methods have largely been used in comparative-effectiveness research assessing the effectiveness of DOACs compared to warfarin [10, 11], only two studies appeared to
compare PS methods or justify PS model selection. In one of these two studies, the use of
IPTW was justified by stating that in survival analysis, PS weighting offers greater bias reduction compared to other methods such as matching or stratification; nevertheless, this was not
empirically tested in their analysis [29].

In the other study carried out by Foslund and colleagues (2018), IPTW and PS by stratification were used in the sensitivity analysis to support the validity of the main Cox regression
analyses, but no direct comparison between methods was made [30].

In addition to this, we have screened eighteen other studies, using PS based methods to
control for confounding, identified from another recent systematic review assessing the comparative- effectiveness of DOACs head-to-head [10]. Among these studies we have identified
only one additional study where different PS approaches were tested [31].

Despite the popularity of PS methods, there are limitations in their application. The generalizability of results may be an important issue when using the matching method as a significant proportion of individuals will be omitted when creating the matched sub-sample. Unlike
the PS matching method, IPW analysis is carried out on the entire cohort. Nevertheless, IPW
offers, along with matching, an important advantage over the covariate adjustment with PS
approach, requiring only the PS model specification for a correct ATE estimation. However,
with poor PS overlap, the resulting extreme weights directly derived from PS may undermine
the robustness of the model [23, 24, 32].

In our analysis, even before adjusting with PS estimates, the baseline characteristics between
groups were already adequately balanced. However, in some cases the standardised differences

indicated that the balance of certain baseline characteristics between treatments did not improve after PS adjustment. This occurrence is reported in the literature, and it seems to be common with the PS matching method when the propensity score is misspecified or matching with replacement is required [25].

Overall patients on dabigatran were younger, with a low risk of stroke and with fewer comorbidities compared to patients on warfarin. This seems to suggest that dabigatran was selectively prescribed to patients with lower risk of stroke and in general healthier than patients on warfarin. Evidence of selective prescribing of dabigatran in younger patients with lower risk of stroke has been reported in the literature [33].

Among the PS methods tested, with a relatively large sample size (DOACs versus warfarin comparison), IPW showed the best covariate balance. However, PS covariate adjustment, less sensitive to sample size not requiring trimming or truncation of extreme weights as with IPW methods, showed the best covariate balance in the dabigatran versus warfarin comparison. Nevertheless, all the different PS methods tested produced treatment effects of similar magnitude. In general, PS covariate adjustment has been perceived as less robust than PS matching and IPW methods, as it is more sensitive to distributional assumptions and PS specification, therefore not reflecting the true treatment effect [23, 24, 32]. Nevertheless, PS covariate adjustment was found to be a valid option to adjust for confounding by indication and in some instances outperformed the other methods reporting much reduced standardised differences.

Moreover, PS methods may not necessarily perform better in assessing covariate imbalance than conventional standard regression. In particular, Elze and colleagues (2017) found that in the presence of substantial covariate imbalance with individuals with very large weights, IPW methods give inaccurate treatment effect estimates. In the case studies evaluated, after truncation, the estimated treatment effect moved towards the crude treatment effect, indicating the inadequacy of these methods in adjusting for covariate imbalance in the presence of heavy weights. On the other hand, the performance of PS matching and standard covariate adjustment were comparable, although PS matching gave less accurate estimates in some instances [34].

## Limitations

In this study we provide an overview of the PS based methods used to address confounding by indication; however, there were a number of limitations inherent to the nature of RWE and PS based methods. Firstly, the relatively small size of the dabigatran subgroup did not allow the analysis to test for PS by stratification, a method involving the stratification of individuals into mutually exclusive subgroups according to their estimated PS [4].

A further constraint in this analysis, concerns the limitation of PS methods of addressing unmeasured confounding which may still bias the estimates. In particular, it is recognised that confounding by indication is the main source of confounding in newly marketed medications where early adopters are most likely to prescribe new drugs when they become available, whereas other prescribers may prefer to opt for existing drugs with proven and established clinical effectiveness. While PS methods can address confounding by indication, there may still be unobserved confounders, such as patients' tolerability and access to healthcare that are difficult to measure [35].

## Conclusion

We have shown how routinely collected linked data can be used to implement PS based methods to generate robust and credible real-world evidence to inform reimbursement and coverage decisions. Propensity score matching and IPW methods are considered theoretically

superior to PS covariate adjustment as the latter may be more prone to model misspecification. In this study, IPW showed the best covariate balance when applied to subgroups with relatively large sample sizes.

However, when applied to subgroups with relatively small sample sizes, using PS as a covariate in the outcome model should be considered, as this method does not require further reduction of sample size, and trimming or truncation of extreme weights. Therefore, relaying on a single method for reducing bias due to confounding should be avoided, as the method of choice may not reflect the true treatment effect, thus leading to an incorrect interpretation of the effect, in the real world, of a given intervention. It follows that, as long as assumptions such as no unobserved confounding hold, several methods should be identified and tested.

As the choice of adequate PS methods may vary according to the characteristics of the observational data available, appropriate methodological design should be in place for comparative-effectiveness analyses including: the assessment of PS overlaps between treatments, inspection of extreme weights, and comparison of PS methods by their standardised difference.

## Supporting information

**S1 Table. Clinical outcomes—ICD-10 and OPCS-4 codes.** ICD:10 = International Statistical Classification of Diseases and Related Health Problems 10th Revision, OPCS-4 = Classification of Interventions and Procedures, BNF = British national formulary, NRS = national records of Scotland.
(PDF)

**S2 Table. Codes for inclusion/exclusion criteria.** ICD:10 = International Statistical Classification of Diseases and Related Health Problems 10th Revision, OPCS-4 = Classification of Interventions and Procedures, BNF = British national formulary, VTE = venous thromboembolism.
(PDF)

## Acknowledgments

We acknowledge the members of the Information Services Division, National Services Scotland, for their vital support.

## Author Contributions

**Conceptualization:** Giorgio Ciminata, Claudia Geue, Olivia Wu, Manuela Deidda, Noemi Kreif, Peter Langhorne.

**Formal analysis:** Giorgio Ciminata.

**Funding acquisition:** Olivia Wu.

**Methodology:** Giorgio Ciminata, Claudia Geue, Olivia Wu, Manuela Deidda, Noemi Kreif, Peter Langhorne.

**Supervision:** Claudia Geue, Olivia Wu, Peter Langhorne.

**Writing – original draft:** Giorgio Ciminata, Claudia Geue, Olivia Wu, Manuela Deidda, Noemi Kreif, Peter Langhorne.

**Writing – review & editing:** Giorgio Ciminata, Claudia Geue, Olivia Wu, Manuela Deidda, Noemi Kreif, Peter Langhorne.

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
