## [Decision Letter · Decision Letter 0]

20 Aug 2021

PONE-D-21-15771

Propensity score methods for comparative-effectiveness analysis: a case study of direct oral anticoagulants in the atrial fibrillation population

PLOS ONE

Dear Dr. Ciminata,

Thank you for submitting your manuscript to PLOS ONE. After careful consideration, we feel that it has merit but does not fully meet PLOS ONE’s publication criteria as it currently stands. Therefore, we invite you to submit a revised version of the manuscript that addresses the points raised during the review process.

We look forward to receiving your revised manuscript.

Kind regards,

Carmine Pizzi

Academic Editor

PLOS ONE

1. Please ensure that your manuscript meets PLOS ONE's style requirements, including those for file naming. The PLOS ONE style templates can be found at https://journals.plos.org/plosone/s/file?id=wjVg/PLOSOne_formatting_sample_main_body.pdf and https://journals.plos.org/plosone/s/file?id=ba62/PLOSOne_formatting_sample_title_authors_affiliations.pdf.

“GC and CG have received research grants from Bristol-Myers Squibb UK and Pfizer UK outside the submitted work. OW has received consulting fee from Bayer UK outside the submitted work. MD, NK and PL declare no conflict of interest.”

**Comments to the Author**

Reviewer #1: The present paper is brilliant and of high qualitative level

Some issues.

In the abstract it should be added how the models performed best, that is which outcomes were evaluated to assess performance of the model.

Methods: why patients in "dabigatran only" were assessed? and not rivaroxaban etc?

Methods: I woudl remore propensity score to mimic rct, which is correct but not formally true

methods dealing with missing value shoud be added

Reviewer #2: The paper is well written and deals with the use of the best methodological approach to estimate the effectiveness of anticoagulation treatment by using a dataset extracted from the Scottish Morbidity Records (SMR). The aim of the authors was to explore the use of Propensity Score (PS) mathcing, Inverse Probability Weighting (IPW) and covariate adjustment with PS. As a clinician, I have appreciated the efforts of the authors to give a clear view on the best statistical analysis to an important topic with crucial influence on Health Technology Assessment (HTA). The Authors have clarified the main differences among the statistical approaches, for example indicating the IPW the best performing method to assess covariate balance when applied to subjects of relatively large sample-size (such as combined DOACs versus warfarin). Instead, covariate adjustment with PS appears to the most appropriate method when applied to subjects of relatively small sample-size (such as dabigatran versus warfarin).

Just one comment on the concept of the introduction of dabigatran in clinical practice. The use of dabigatran as opposed to warfarin is more linked to the more convenient management of the drug than to other reasons. This is what has happened in the recents years in the clinical practice, but the same also is seen for apixaban and rivaroxaban.

---

## [Author Response · Author response to Decision Letter 0]

13 Oct 2021

We would like to thank both reviewers for their time and valuable comments. We have addressed these as follows.

Reviewer#1: In the abstract it should be added how the models performed best, that is which outcomes were evaluated to assess performance of the model.

Response: In the Abstract the following is already stated: “Model performance was assessed by standardised difference. Clinical outcomes and mortality were compared for all DOACs (including apixaban, dabigatran and rivaroxaban) versus warfarin”. The outcomes used in our model are now specified in the methods section of the abstract (line 41). However, the clinical outcomes where not used to assess model performance, but for indicating how the size of the treatment effect (reported in the form of Hazard Ratios) may change according to the PS method employed. For instance, with the IPTW-IPCW approach, the treatment effect tends to be larger, but still in line, compared to the treatment effect estimated with other PS methods. This is now stated in the results section of the Abstract (line 48 and 49). 

Reviewer#1: why patients in "dabigatran only" were assessed? and not rivaroxaban etc?

Response: An explanation on why patients on “dabigatran only” were assessed is already provided in the manuscript; that is the smallest subgroup of DOACs users in Scotland compared to apixaban and rivaroxaban. However, this has now been made clearer (line 169). 

Reviewer#1: Methods: I would remove propensity score to mimic RCT, which is correct but not formally true.

Response: The sentence referring to propensity score mimicking RCT has now been deleted (line 174).

Reviewer#1: methods dealing with missing value should be added.

Response: No methods for dealing with missing value were used. Given that missing data was <5%, no imputation method was used, a complete available case analysis was provided instead. This is now stated in the manuscript (line 194-196).

Reviewer #2: Just one comment on the concept of the introduction of dabigatran in clinical practice. The use of dabigatran as opposed to warfarin is more linked to the more convenient management of the drug than to other reasons. This is what has happened in the recent years in the clinical practice, but the same also is seen for apixaban and rivaroxaban.

Response: More on the “more convenient management” of DOACs versus warfarin has now been added to the methods section (line 110-13). That is, rapid onset of action, and the possibility of using fixed doses without requirement for routine anticoagulant monitoring. 

Kind Regards,

Dr Giorgio Ciminata

---

## [Decision Letter · Decision Letter 1]

21 Dec 2021

Propensity score methods for comparative-effectiveness analysis: a case study of direct oral anticoagulants in the atrial fibrillation population

PONE-D-21-15771R1

Dear Dr. Ciminata,

We’re pleased to inform you that your manuscript has been judged scientifically suitable for publication and will be formally accepted for publication once it meets all outstanding technical requirements.

Kind regards,

Carmine Pizzi

Academic Editor

PLOS ONE

---

## [Editor Report · Acceptance letter]

12 Jan 2022

PONE-D-21-15771R1 

Propensity score methods for comparative-effectiveness analysis: a case study of direct oral anticoagulants in the atrial fibrillation population 

Dear Dr. Ciminata:

I'm pleased to inform you that your manuscript has been deemed suitable for publication in PLOS ONE. Congratulations! Your manuscript is now with our production department. 

Kind regards, 

on behalf of

Prof Carmine Pizzi 

Academic Editor

PLOS ONE